# Adversarial EXEmples: Functionality-preserving Optimization of Adversarial Windows Malware

Luca Demetrio [1]   Battista Biggio [1 2]   Giovanni Lagorio [3]   Alessandro Armando [3]   Fabio Roli [1 2]

## Abstract

Windows malware classifiers that rely on static analysis have been proven vulnerable to adversarial EXEmples, i.e., malware samples carefully manipulated to evade detection. However, such attacks are typically optimized via query-inefficient algorithms that iteratively apply random manipulations on the input malware, and require checking that the malicious functionality is preserved after manipulation through computationally-expensive validations. To overcome these limitations, we propose *RAMEn*, a general framework for creating adversarial EXEmples via functionality-preserving manipulations. RAMEn optimizes their parameters of such manipulations via gradient-based (white-box) and gradient-free (black-box) attacks, implementing many state-of-the-art attacks for crafting adversarial Windows malware. It also includes a family of black-box attacks, called GAMMA, which optimize the injection of benign content to facilitate evasion. Our experiments show that gradient-based and gradient-free attacks can bypass malware detectors based on deep learning, non-differentiable models trained on hand-crafted features, and even some renowned commercial products.

## 1. Introduction

Windows malware is still a threat in the wild, as thousands of malicious programs are uploaded to VirusTotal every day.[1] To counter such trend, modern approaches use machine learning to detect these threats at scale (Saxe & Berlin, 2015; Kolosnjaji et al., 2016; Hardy et al., 2016; David & Netanyahu, 2015; Incer et al., 2018; Anderson & Roth, 2018; Raff et al., 2018).

However, these techniques have not been tested under the lens of *adversarial machine learning* (Huang et al., 2011; Biggio & Roli, 2018), that studies the security aspects of machine-learning algorithms under attacks staged either at training or at test time. In particular, recent work has shown how an attacker can create *adversarial EXEmples*, i.e., Windows malware samples carefully perturbed to evade learning-based detection while preserving malicious functionality (Kolosnjaji et al., 2018; Demetrio et al., 2019; 2021; Anderson et al., 2017; Castro et al., 2019a; Kreuk et al., 2018; Sharif et al., 2019). Manipulating programs is not as easy as perturbing images, hence the attacker can either apply invasive perturbations and use sandboxes to ensure that functionality of the binary is not compromised (Castro et al., 2019a; Song et al., 2020), wasting a lot of queries to the target model in the process, or focus the perturbation on areas of the file that do not impact functionality (Kreuk et al., 2018; Demetrio et al., 2019; Kolosnjaji et al., 2018).

To overcome these limitations, we propose *RAMEn* as an unifying framework for creating adversarial EXEmples by leveraging *practical* manipulations, i.e. transformations that alter the representation of a program without compromising its original functionality. This allows the attacker to skip the validation inside a sandbox, saving computations and queries sent to the target. This framework is formalized as a minimization problem over the parameters of such manipulations, and it can be optimized with both gradient-based (white-box) and gradient-free (black-box) techniques. We test both scenarios, showing that end-to-end networks are weak against minimal perturbations, and that decision trees trained on hand-crafted features can be bypassed by injecting content harvested from legitimate goodware samples. We also remark how the latter can transfer also against commercial products on VirusTotal [2], successfully bypassing 12 of them.

*Equal contribution [1]Dipartimento di Ingegneria Elettrica ed Elettronica, Università degli Studi di Cagliari, Italy [2]Pluribus One, Italy [3]Dipartimento di Informatica, Bioingegneria, Robotica e Ingegneria dei Sistemi, Università degli Studi di Genova, Italy. Correspondence to: Luca Demetrio <luca.demetrio93@unica.it>.

*Accepted by the ICML 2021 workshop on A Blessing in Disguise: The Prospects and Perils of Adversarial Machine Learning.* Copyright 2021 by the author(s).

[1]https://www.virustotal.com/it/statistics/

[2]https://virustotal.com

## 2. Formalization and Manipulations

Let be $\mathcal{Z} \subset \{0,\dots,255\}^*$ all possible functioning programs in the input space as string of bytes. We define the target detector as two different components: the feature extractor $\phi : \mathcal{Z} \to \mathcal{X}$, being $\mathcal{X} \subseteq \mathbb{R}^d$ a $d$-dimensional vector space (i.e., the feature space); and the the prediction function $f : \mathcal{X} \to \mathbb{R}$. On top of these objects, we formalize *RAMEn*, a general framework that reduces the problem of computing adversarial EXEmples to optimization problems of the form:

$$\underset{\boldsymbol{t} \in \mathcal{T}}{\text{minimize}} \quad F(\boldsymbol{t}) = L(f(\phi(h(\boldsymbol{z}, \boldsymbol{t})), y) . \quad (1)$$

where $L : \mathbb{R} \times \mathcal{Y} \to \mathbb{R}$ is a *loss function* that measures how likely an input sample is classified as malware, by comparing the output of the prediction $f(\phi(\boldsymbol{z}))$ on a malicious input sample $\boldsymbol{z}$ against the class label $y = -1$ of benign samples. This problem is connected to the *practical* manipulations (Demetrio et al., 2020), formalized as a function $h : \mathcal{Z} \times \mathcal{T} \to \mathcal{Z}$. These are functions that perturb malware samples without compromising their original functionality, allowing the attacker to skip any validation step, speeding up the procedure and saving resources.

**Practical manipulations.** Malware can be manipulated by using two different families of manipulation functions, either by altering its file representation, or by altering its code. We focus on the first family, since we are interested in testing the robustness of detectors that work at static time, and we report the manipulations that address the runtime of a program in the Appendix. To alter the representation, the attacker leverage ambiguities of the format used for storing program as regular file, i.e. the Windows PE file format. [3] We leave to the Appendix a more detailed overview of such format.

*(s.1) Perturb Header Fields* (Anderson et al., 2017; Castro et al., 2019a;b). This technique includes altering section names, breaking the checksum, and altering debug information.

*(s.2) Filling Slack Space* (Kreuk et al., 2018; Anderson et al., 2017; Castro et al., 2019b;a). This technique manipulates the slack space inserted by the compiler to maintain the alignments inside the file. The corresponding slack bytes are usually set to zero, and they are never referenced by the code of the executable.

*(s.3) Padding* (Kolosnjaji et al., 2018; Kreuk et al., 2018). This technique injects additional bytes at the end of the file.

*(s.4) Manipulating DOS Header and Stub* (Demetrio et al., 2019; 2020). This technique edit partially or completely the DOS Header and Stub of a program, which are not used by modern programs.

*(s.5) Extend the DOS Header* (Demetrio et al., 2020). This technique extends the DOS header by injecting content be-

---

[3] https://docs.microsoft.com/it-it/windows/win32/debug/pe-format

---

**Algorithm 1** Gradient-based attack for optimizing Eq. 1

**Input** : $\boldsymbol{z}$, initial malware sample; $N$, iterations; $y$, target class label; $f$, target model.
**Output** : $\boldsymbol{z}^\star$, the adversarial EXEmple.
1  $\boldsymbol{t}^{(0)} \in \mathcal{T}$
2  **for** $i$ **in** $[0, N-1]$ **do**
3  $\quad \boldsymbol{x}' \leftarrow \phi(h(\boldsymbol{z}, \boldsymbol{t}^{(i)}))$
4  $\quad \boldsymbol{x}^\star \leftarrow \arg \min_{\boldsymbol{x}' \in \mathcal{X}} L(f(\boldsymbol{x}'), y)$
5  $\quad \boldsymbol{t}^{(i+1)} \leftarrow \arg \min_{\boldsymbol{t} \in \mathcal{T}} \|\boldsymbol{x}^\star - \phi(h(\boldsymbol{z}, \boldsymbol{t}^{(i)}))\|^2$
6  $\boldsymbol{z}^\star \leftarrow h(\boldsymbol{z}, \boldsymbol{t}^{(N)})$
7  return $\boldsymbol{z}^\star$

---

**Algorithm 2** Gradient-free (Black-box) Attacks for Optimizing Adversarial Malware EXEmples in RAMEn.

**Data:** $\boldsymbol{z}$, initial malware sample; $N$, total number of iterations; $y$, target class label; $f$, target model function
**Result:** $\boldsymbol{z}^\star$, the adversarial EXEmple.
8  $\boldsymbol{t}^{(0)} \in \mathcal{T}$
9  **for** $i$ **in** $[0, N-1]$ **do**
10  $\quad \boldsymbol{t}^{(i+1)} = \arg \min_{\boldsymbol{t} \in \mathcal{T}} L(f(\phi(h(\boldsymbol{z}, \boldsymbol{t}^{(i)}))), y)$
11  $\boldsymbol{z}^\star = h(\boldsymbol{z}, \boldsymbol{t}^{(N)})$
12  return $\boldsymbol{z}^\star$

---

fore the actual header of the program.

*(s.6) Content shifting* (Demetrio et al., 2020). This technique creates additional space before the beginning of a section, by shifting the content forward, and injects adversarial content in between.

*(s.7) Import Function Injection* (Anderson et al., 2017; Castro et al., 2019a;b). This technique injects import functions by adding an appropriate entry to the Import Address Table, specifying which function from which library must be included during the loading process.

*(s.8) Section Injection* (Anderson et al., 2017; Castro et al., 2019a;b). This technique injects new sections into the input file by creating an additional entry inside the section table. Each section entry is 40 bytes long, so all the content has to be shifted by that amount, without compromising file and section alignments as specified by the header.

**Solving the minimization.** Depending on the differentiability of the terms that build Eq. 1, the attacker can decide to land gradient-based (white-box) or gradient-free (black-box) attacks.

*Gradient-based attacks.* In the case of end-to-end security detectors, the model function $f$ is differentiable, but both the feature extractor $\phi$ and the manipulation $h$ are not. To overcome this issue, the proposed attacks perform gradient descent in the feature space, while iteratively trying to reconstruct the corresponding adversarial malware example in the input space, using different strategies, as shown in Alg. 1. The procedure firstly perturb the sample, and

it encodes it into the feature space (line 3). It then solves the problem in feature space, by applying a gradient descent algorithm on the loss function $L$ (line 4). Since the attacker wants a real functioning malware, they must invert the feature mapping, hence solving such reconstruction as a minimization problem, where the output is the best vector $t$ that creates an adversarial EXEmple as close as possible to the one computed in feature space (line 5). This strategy is general enough for recasting most of the proposed attacks inside RAMEn, as shown in Table 1, by defining the building blocks of this procedure: the *loss function* to be minimized, the *practical manipulations* they will use, the algorithm for the *feature-space optimization*, and procedure for *reconstructing* the sample in input space.

*Gradient-free attacks.* Either when the target model is unavailable, or it is not-differentiable, the attacker must rely on gradient-free techniques, as described by Algorithm 2. In each iteration, the attack solves the minimization problem with the chosen optimizer, perturbing the malware with the given practical manipulations (line 10). Again, this procedure is general enough for recasting already-proposed techniques, and we show such generalization in Table 2. To do so, the attacker must define the *loss function* to be minimized, the *practical manipulations* they will use, and the black-box optimizer that will combine them together. To speed up the optimization and save more queries, we propose GAMMA, a black-box optimizer that solves Eq. 1 by using practical manipulations that injects content taken from goodware samples (Demetrio et al., 2021). In this way, the optimizer does not have to explore blindly the space of solutions, but rather aggregating patterns of byte that decrease most the loss of the attack. Such loss is also enriched with a regularization term $C$ that weights the size of the resulting adversarial EXEmple, controlled by a parameter $\lambda$.

## 3. Experimental results

We now test attacks encoded using RAMEn against state-of-the-art classifiers.

**Gradient-based attacks.** We select two architectures as target of our gradient-based attacks, both of them take as input an unprocessed malware, and they compute a malicious score by looking at their bytes. One is *MalConv* (Raff et al., 2018), that uses the first 1 MB of the input sample, and the other are *DNN-Lin and DNN-ReLU* (Coull & Gardner, 2019) depending on the activation function that has been used, both using the first 100 KB of the input sample as input. For both architectures, each sample is padded with a special character or truncated if necessary. All networks are trained using the EMBER (Anderson & Roth, 2018) dataset, and also we test the robustness of DNN-Lin and DNN-ReLU trained on a proprietary dataset of 16M samples (Demetrio et al., 2020). We test these networks using

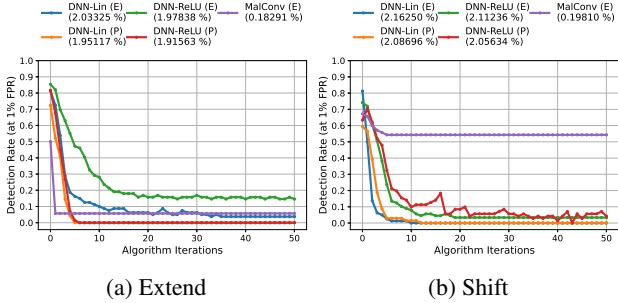

| (a) Extend | (b) Shift |
|---|---|

*Figure 1.* Gradient-based attacks against end-to-end networks. The numbers below each label represent the percentage of the input window length that has been manipulated.

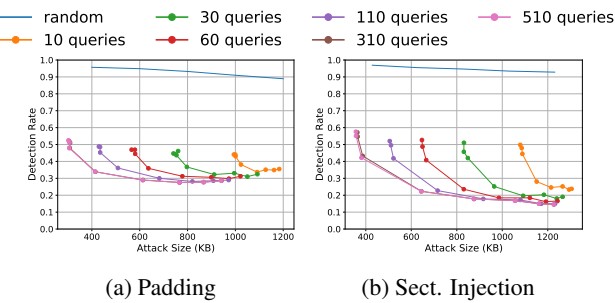

| (a) Padding | (b) Sect. Injection |
|---|---|

*Figure 2.* Gradient-free GAMMA attacks against the Gradient Boosting Decision Tree (GBDT). Each curve represent a particular combination of query budget and regularization parameter.

our gradient optimizer (Demetrio et al., 2021), and we show the efficacy of manipulating malware using *s.5* and *s.6*. We also report the results of other practical manipulations in the Appendix, as the chosen two outperforms the other in terms of decrement of the detection rate. The optimizer run for 50 iterations, and at each step, the algorithm perturbs at most 256 bytes inside the sample. We show the results of our attacks in Fig. 1, where we plot the decrement of the detection rate while optimizing the attack. The detection threshold of each classifier has been set to match their performance at 1% False Positive Rate (FPR). Both the *Extend* and *Shift* attacks replace a portion of the real header of the program, and it might be possible that the adversarial noise interferes with the local patterns learned by the networks at training time, i.e. the position of the meaningful metadata of the program. **Gradient-free attacks** We select a state-of-the-art gradient boosting decision tree (GBDT) trained on hand-crafted features (Anderson & Roth, 2018), extracted from the EMBER dataset. We consider GAMMA as attacking algorithm, and we use *padding (s.3)* and *section injection (s.8)* as practical manipulations. We extract 75 `.rdata` sections from goodware program to use as content to be injected using both the *section injection* and *padding* manipulations. we set different query budgets $T$ between 10 and 510, and regularization parameters $\lambda \in \{10^{-i}\}_{i=3}^{9}$. We plot the efficacy of GAMMA in Fig. 2, where we report

| Attack | Loss Function $L$ | Practical Manipulations $h(\cdot, \boldsymbol{t})$ | Feature-space Optimization | Input-space Reconstruction |
|---|---|---|---|---|
| Full DOS (Demetrio et al., 2020) | malware score | manipulate all DOS header | single gradient step | closest positive (iterative) |
| Extend (Demetrio et al., 2020) | malware score | extend DOS header | single gradient step | closest positive (iterative) |
| Shift (Demetrio et al., 2020) | malware score | shift section content | single gradient step | closest positive (iterative) |
| Padding (Kolosnjaji et al., 2018) | malware score | padding | single gradient step | closest positive (iterative) |
| Partial DOS (Demetrio et al., 2019) | malware score | partial DOS header | single gradient step | closest positive (iterative) |
| FGSM (Kreuk et al., 2018) | malware score | padding + slack space | FGSM | closest (non-iterative) |
| Binary Diversification (Sharif et al., 2019) | CW loss | equivalent instructions | single gradient step | gradient-aligned transformation (iterative) |

*Table 1.* Recasting gradient-based (white-box) attacks within RAMEn, according to the steps detailed in Algorithm 1.

| Attack | Loss Function $L$ | Practical Manipulations $h(\cdot, \boldsymbol{t})$ | Optimizer | Validation |
|---|---|---|---|---|
| Full DOS (Demetrio et al., 2020) | malware score | manipulate all DOS header | genetic | none |
| Extend (Demetrio et al., 2020) | malware score | extend DOS header | genetic | none |
| Shift (Demetrio et al., 2020) | malware score | shift section content | genetic | none |
| Partial DOS (Demetrio et al., 2019) | malware score | partial DOS header | genetic | none |
| Padding (Kolosnjaji et al., 2018) | malware score | padding | genetic | none |
| GAMMA (Demetrio et al., 2021) | malware score + size penalty | padding with benign sections / benign section injection | genetic | none |
| RL Agent (Anderson et al., 2017) | malware score | padding + section / API inj. + header fields + binary rewriting | reinforcement learning | none |
| AIMED (Castro et al., 2019a) | malware score | padding + section/API inj. + header fields + binary rewriting | genetic | sandbox |
| AEG (Song et al., 2020) | malware score | padding + section inj. + header fields + binary rewriting | random manipulations | sandbox |

*Table 2.* Recasting gradient-free (black-box) attacks within RAMEn, according to the steps detailed in Algorithm 2

the efficacy of the usage of *Padding* (Fig. 2a) and *Section Injection* (Fig. 2b). As the value of $\lambda$ decreases, the algorithm finds more evasive samples with bigger payloads, since the penalty term is negligible while computing the objective function. On the other hand, by increasing the value of $\lambda$ the resulting attack feature vector become sparse, generating smaller but more detectable adversarial example. In this case, the penalty term engulfs the score computed by the classifier, which becomes irrelevant during the optimization. Also, the query budget matters, since GAMMA can explore more solutions that are stealthy and evasive at the same time, but such solutions could not be found at early stages of the optimization process. To prove the efficacy of our methodology, we report the results of the application of random byte sequences of increasing length. This experiment highlights a slight descending trend, but the optimized attack with benign content injection is way more effective than random perturbations. The detection rate of GBDT is decreased more by the section-injection attack than by padding. Since the first technique also introduces a section entry inside the section table, the adversarial payload perturbs more features than those modified by the padding attack.

**Transfer attack on VirusTotal.** We test the robustness of commercial products leveraging responses from VirusTotal,[4] an online interface for many threat detectors. We use the results obtained against the GBDT classifier through the application of the *Section Injection* manipulation, compared to a baseline random attack that injects 50 KB of random content. We report in Table 3 the detection rate of the commercial products hosted on VirusTotal (70 in total). While the random attack only slightly decreases the number of detections per sample, the section-injection attack is able

|  | **Malware** | **Random** | **Sect. Injection** |
|---|---|---|---|
| **Detections** | $46 \pm 12$ | $41 \pm 12$ | $34 \pm 13$ |

*Table 3.* Detection achieved by commercial products hosted on VirusTotal, before and after manipulations.

to bypass an average of more than 12 detectors per sample. We report less aggregated results inside the Appendix. The reason may be that some of these antivirus programs already use static machine learning-based detectors to implement a first line of defense when protecting end-point clients from malware, as also confirmed in their blog or website, and this makes them more vulnerable to our attacks.

## 4. Conclusions

We propose RAMEn, a lightweight formalization for computing adversarial EXEmples, leveraging practical manipulations that do no alter the original functionality of input samples, and general enough for encoding both gradient-based and gradient-free approaches. We show that we can recast most of the proposed attacks inside RAMEn, and we parametrize the manipulations to allow the optimization of the injected content. We analyze the efficacy of both gradient-based and gradient-free attacks, highlighting the weakness of both end-to-end deep networks and a detector based on hand-crafted features. The latter is also used for computing transfer attacks against commercial product, successfully evading 12 of them. As future work, we would like either to extend our attacks to detectors that extract features from the execution of malware, or also to improve the robustness of machine learning malware classifiers by leveraging domain knowledge in the form of constraints and regularizers.

---

[4] https://virustotal.com

## Acknowledgement

This work was partly supported by the PRIN 2017 project RexLearn (grant no. 2017TWNMH2), funded by the Italian Ministry of Education, University and Research; and by the EU H2020 project ALOHA, under the European Union's Horizon 2020 research and innovation programme (grant no. 780788).

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

# Appendix

## A. Windows PE File format

The *Windows Portable Executable* (PE)[5] format specifies how programs are stored as a file on disk, and it instructs the loader on where to find the software main components.

**DOS Header and Stub.** The DOS header contains metadata for loading the executable inside a DOS environment, while the DOS Stub is made up of few instructions that will print "*This program cannot be run in DOS mode*" if executed inside a DOS environment. These two components have been kept to maintain compatibility with older Microsoft operating systems.

**PE Header.** The real header of the program contains the magic number PE and the characteristics of the executable, i.e. the target architecture , the size of the header and attributes of the file.

**Optional Header.** It contains the information needed by the OS for loading the binary into memory such as: (i)*file alignment*, that acts as a constraint on the structure of the executable since each section of the program must start at an offset multiple to that field, and the (ii) *size of headers* that specifies the amount of bytes that are reserved to all the headers of the programs, and it must be a multiple of the *file alignment*. Lastly, this header contains offsets that point to other structures, like the Import Table (needed by the OS for resolving dependencies), the Export Table (to find functions that can be referenced by other programs), and more.

**Section Table.** It is a list of entries that indicates the characteristics of each section of the program. Each entry is provided with a name, an offset to the location inside the binary, a virtual address where the content should be mapped in memory, and the characteristics of such content (i.e. is read-only, write-only, or it is executable, and more).

**Sections.** These are contiguous chunks of bytes, loaded in memory by the loader after the parsing of the Section Table. To maintain the alignment specified inside the Optional Header, these sections might be zero-padded to match the format constraint.

## B. Behavioral manipulations.

These functions alter the code of input programs, by either replacing instructions, adding new ones, or by applying packing techniques (Wenzl et al., 2019).
*(b.1) Packing* (Anderson et al., 2017; Castro et al., 2019a;b). This technique amounts to encrypting or encoding the content of the binary inside another binary and decoding it at run-time. The effect of a packer is invasive since the whole structure of the input sample is modified.
*(b.2) Direct* (Wenzl et al., 2019). This approach rewrites specific portions of the code, like replacing assembly instructions with equivalent ones (e.g., additions and subtractions with opposed sign).
*(b.3) Minimal Invasive* (Anderson et al., 2017; Wenzl et al., 2019). This technique sets the entry-point to a new executable section that jumps back to the original code.
*(b.4) Full Translation* (Wenzl et al., 2019). This approach lifts all the code to a higher representation, e.g., LLVM,[6] since it simplifies the application of perturbations, and it then translates the code back to the assembly language.
*(b.5) Dropper* (Ceschin et al., 2019). This approach stores the code as a resource of another binary, which is then loaded at runtime.

## C. Gradient-based attacks

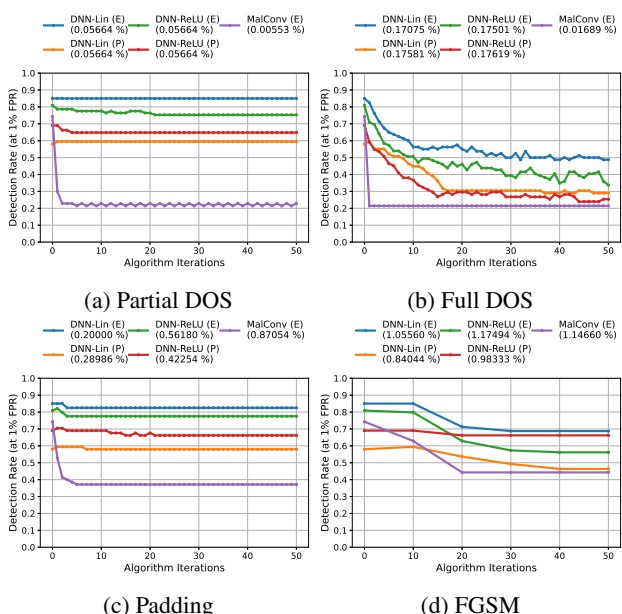

(a) Partial DOS  (b) Full DOS

(c) Padding  (d) FGSM

*Figure 3.* Gradient-based attacks using practical manipulations, against end-to-end neural networks.

---

[5]https://docs.microsoft.com/en-us/windows/win32/debug/pe-format

[6]https://llvm.org/

|     | Malware | Random | Sect. Injection |
|-----|---------|--------|-----------------|
| AV1 | 93.5%   | 85.5%  | 30.5%           |
| AV2 | 85.0%   | 78.0%  | 68.0%           |
| AV3 | 85.0%   | 46.0%  | 43.5%           |
| AV4 | 84.0%   | 83.5%  | 63.0%           |
| AV5 | 83.5%   | 79.0%  | 73.0%           |
| AV6 | 83.5%   | 82.5%  | 69.5%           |
| AV7 | 83.5%   | 54.5%  | 52.5%           |
| AV8 | 76.5%   | 71.5%  | 60.5%           |
| AV9 | 67.0%   | 54.5%  | 16.5%           |

*Table 4.* Detection rate of 9 antivirus programs from VirusTotal computed on ($i$) the initial set of 200 malware samples, and on the same samples manipulated with ($ii$) random attacks and ($iii$) section-injection attacks.

We show in Fig. 3 the efficacy of other strategies proposed in literature. We apply the manipulation $s.4$, editing either a fraction of the DOS header (*Partial DOS*, Fig. 3a) or all of it (*Full DOS*, Fig. 3b). While the *Partial DOS* technique is generally ineffective against all classifiers except for Mal-Conv (as already pointed put by Demetrio et al. (Demetrio et al., 2019)), the *Full DOS* attack does substantially lower the detection rate of the networks proposed by Coull et al. (Coull & Gardner, 2019). This might be caused by spurious correlations learnt by the network, and altering these values cause the classifier to lose precision. We test the $s.3$ manipulation (*Padding*, Fig. 3c) (Kolosnjaji et al., 2018), and the *FGSM* attack proposed by Kreuk et al. (Kreuk et al., 2018) (Fig. 3d) do not decrease much the detection rates of the networks, since most of the manipulations applied are cut off by the limited window size of the network itself. For instance, if a sample is larger than `100` KB, it can not be padded, and all the strategies that rely on padding fail. To achieve evasion, these FGSM attacks can only leverage the perturbation of the slack space, but the number of bytes that can be safely manipulated is too few to have significant impact. Also, this strategy is incapacitated by the inverse-mapping problem: they compute the adversarial examples inside the feature space, and they project them back only at the end of the algorithm. This means that the attack might be successful inside the feature space, but not inside the input space, where there are a lot of constraints that are ignored by the attack itself. Against MalConv, the *Padding* attack proves to be quite effective, but it needs at most 10 KB to land successful attacks, as already highlighted by Kolosnjaji et al. (Kolosnjaji et al., 2018). The adversarial payload must include as many bytes as possible to counterbalance the high score carried by the ones contained inside the header.

## D. Detailed Detection Rate of Commercial Products

To better highlight such result, we report on Table 4 the detection rates of 9 different antivirus products that appear on the 2019 Gartner Magic Quadrant for Endpoint Protection Platforms,[7] including many leading and visionary products, before and after executing the random and section-injection attacks. In many cases, our section-injection attack is able to drastically decrease the detection rate (e.g., AV1, AV3, AV7 and AV9), significantly outperforming the random attack (e.g., AV1 and AV9).

---

[7]https://www.microsoft.com/security/blog/2019/08/23/gartner-names-microsoft-a-leader-in-2019-endpoint-protection-platforms-magic-quadrant/