# OpenReview forum: "Adversarial EXEmples: Functionality-preserving Optimization of Adversarial Windows Malware"
_ICML.cc/2021/Workshop/AML — ICML 2021 Workshop AML Poster_

### Official Review · Reviewer_pY4c · 2021-06-20
**A preliminary work applying adversarial machine learning to modern machine learning based Windows malware detectors**

**Rating:** Accept
**Confidence:** 3

**Review:**

The authors formalize the problem of computing adversarial EXEmples via functionality preserving manipulations as an optimization problem, which involves a feature extractor $\phi$, a prediction function $f$, a loss function $L$, and a function $h$ to represent possible functionality-preserving manipulations. Thus the authors propose a unified lightweight framework $-$ RAMEn. This framework is quite general and can recast many existing works. The authors carried out experiments to show its effectiveness.

The paper is clear in general. However, more backgrounds might be helpful. For example, providing more backgrounds on the layout of the Windows PE file format might help to understand why the s.1 to s.8 manipulations do not change the functionality of the program.

Pros:
1. The authors propose a unified framework for both gradient-based and gradient-free threat models. Many existing works can be recasted within it.
2. The transfer-attack experiment demonstrates that the proposed approach can be used to compute adversarial EXEmples for real-world commercial products.

Cons:
1. The insight of using functionality-preserving manipulations is used by many existing works. The s.1 to s.8 manipulations are proposed by existing works, too.
2. Some details of the experiments are not well-explained, e.g., the reason for choosing the s.5 and s.6 manipulations for the gradient-based attacks experiment is not explained.

---

### Decision · Program_Chairs · 2021-06-21

**Decision:**

Accept (Poster)

**Comment:**

This paper studied adversarial attacks on machine learning based Windows malware detectors. The paper is well-written. The authors can further address the reviewer's comments.